# Phosphorous Magnetic Resonance Spectroscopy and Molecular Markers in IDH1 Wild Type Glioblastoma

**DOI:** 10.3390/cancers13143569

**Published:** 2021-07-16

**Authors:** Malik Galijašević, Ruth Steiger, Ivan Radović, Anna Maria Birkl-Toeglhofer, Christoph Birkl, Lukas Deeg, Stephanie Mangesius, Andreas Rietzler, Milovan Regodić, Guenther Stockhammer, Christian Franz Freyschlag, Johannes Kerschbaumer, Johannes Haybaeck, Astrid Ellen Grams, Elke Ruth Gizewski

**Affiliations:** 1Department of Neuroradiology, Medical University of Innsbruck, 6020 Innsbruck, Austria; malik.galijasevic@i-med.ac.at (M.G.); ivan.radovic@student.i-med.ac.at (I.R.); christoph.birkl@i-med.ac.at (C.B.); lukbox@gmx.de (L.D.); stephanie.mangesius@i-med.ac.at (S.M.); andreas.rietzler@i-med.ac.at (A.R.); astrid.grams@i-med.ac.at (A.E.G.); elke.gizewski@i-med.ac.at (E.R.G.); 2Neuroimaging Research Core Facility, Medical University of Innsbruck, 6020 Innsbruck, Austria; 3Institute of Pathology, Neuropathology and Molecular Pathology, Medical University of Innsbruck, 6020 Innsbruck, Austria; anna.birkl-toeglhofer@i-med.ac.at (A.M.B.-T.); johannes.haybaeck@i-med.ac.at (J.H.); 4Department of Otorhinolaryngology, Medical University of Innsbruck, 6020 Innsbruck, Austria; milovan.regodic@student.i-med.ac.at; 5Department of Radiation Oncology, Medical University of Vienna, 1010 Vienna, Austria; 6Department of Neurology, Medical University of Innsbruck, 6020 Innsbruck, Austria; guenther.stockhammer@i-med.ac.at; 7Department of Neurosurgery, Medical University of Innsbruck, 6020 Innsbruck, Austria; Christian.Freyschlag@i-med.ac.at (C.F.F.); johannes.kerschbaumer@i-med.ac.at (J.K.); 8Diagnostic and Research Center for Molecular Biomedicine, Institute of Pathology, Medical University of Graz, 8010 Graz, Austria

**Keywords:** glioblastoma, phosphorous spectroscopy, brain energy metabolism

## Abstract

**Simple Summary:**

Gliobastoma is one of the deadliest tumors overall, yet the most common malignant brain tumor. The new World Health Organization Classification of Brain Tumors brought changes in how we look at this type of malignancy. Now we know that glioblastoma is rather a spectrum of similar tumors, but with some distinct characteristics that include molecular footprint, response to therapy and with that overall survival, among others. We hypothesised that by employing phosphorous magnetic resonance we will be able to show differences in cellular energy metabolism in these various subtypes of glioblastoma. For example, we found indices of faster cell reproduction and tumor growth in MGMT-methylated and EGFR-amplified tumors. These tumors also could have reduced energetic state or tissue oxygenation due to the increased necrosis. Tumors with EGFR-amplification could have increased apoptotic activity regardless of their MGMT status. Our study indicated various differences in energetic metabolism in tumors with different molecular characteristics, which could potentially be important in future therapeutic strategies.

**Abstract:**

The World Health Organisation’s (WHO) classification of brain tumors requires consideration of both histological appearance and molecular characteristics. Possible differences in brain energy metabolism could be important in designing future therapeutic strategies. Forty-three patients with primary, isocitrate dehydrogenase 1 (IDH1) wild type glioblastomas (GBMs) were included in this study. Pre-operative standard MRI was obtained with additional phosphorous magnetic resonance spectroscopy (31-P-MRS) imaging. Following microsurgical resection of the tumors, biopsy specimens underwent neuropathological diagnostics including standard molecular diagnosis. The spectroscopy results were correlated with epidermal growth factor (EGFR) and $O^{6}$-Methylguanine-DNA methyltransferase (MGMT) status. EGFR amplified tumors had significantly lower phosphocreatine (PCr) to adenosine triphosphate (ATP)-PCr/ATP and PCr to inorganic phosphate (Pi)-PCr/Pi ratios, and higher Pi/ATP and phosphomonoesters (PME) to phosphodiesters (PDE)-PME/PDE ratio than those without the amplification. Patients with MGMT-methylated tumors had significantly higher cerebral magnesium (Mg) values and PME/PDE ratio, while their PCr/ATP and PCr/Pi ratios were lower than in patients without the methylation. In survival analysis, not-EGFR-amplified, MGMT-methylated GBMs showed the longest survival. This group had lower PCr/Pi ratio when compared to MGMT-methylated, EGFR-amplified group. PCr/Pi ratio was lower also when compared to the MGMT-unmethylated, EGFR not-amplified group, while PCr/ATP ratio was lower than all other examined groups. Differences in energy metabolism in various molecular subtypes of wild-type-GBMs could be important information in future precision medicine approach.

## 1. Introduction

The recent update of the WHO classification of brain tumors brought significant changes in how we look at the neuropathology of gliomas. Molecular status of the tumor is a crucial part of the diagnosis, together with the histological appearance.

Grade IV gliomas are now divided in glioblastomas (GBMs) and H3K27M mutant diffuse midline gliomas. GBM is the most devastating, yet most common of all primary brain intrinsic tumors. Even with the best possible medical care, 5-year-survival is limited to 5% [1]. The Cancer Genome Atlas (TCGA) Research Group analysed over 500 GBM samples and published a “genomic landscape” of GBM [2]. Recently, the World Health Organisation (WHO) included molecular parameters in defining final diagnosis of tumors. GBM is divided into:infrequent isocitrate dehydrogenase (IDH)-mutant GBM, which is actually secondary in nature [3]–derived from a lower grade glioma (<10% of cases),IDH1-wild type (IDHwt) or primary GBM, andGBM-not otherwise specified (GBM NOS).

Furthermore, IDH wild type GBM subtypes are giant cell GBM, gliosarcoma, and epitheloid GBM [4]. H3K27M mutant diffuse midline glioma is a newly termed tumor type found mostly in children and young adults [4,5].

O6-Methylguanin-DNA-Methyltransferase (MGMT) promoter methylation status was shown to be the major marker for prognosis and treatment response in GBM [6]. MGMT is a nuclear protein involved in repair of alkylated DNA. Due to this characteristics, patients with methylated MGMT GBMs respond better to the therapy with alkylated agents such as temozolomide [7,8].

The status of the epidermal growth factor receptor (EGFR) should also be routinely assessed. EGFR is a cell membrane protein that acts as a receptor for epidermal growth factor, regulating cellular proliferation, differentiation, and survival [9].

According to the joint Austrian, German, and Swiss guidelines for gliomas in adults [5], contrast enhanced magnetic resonance imaging (MRI) remains the method of choice for imaging brain tumors.

Phosphorous magnetic resonance spectroscopy is an MRI technique used to depict levels of various phosphorous metabolites in vivo. Phosphates, compounds that contain the phosphate ($PO_{4}$) are extremely important and abundant in human tissues. Only stable isotope of phosphorous, ${}^{31}P$ has a non zero spin and can therefore undergo resonance. With 31-P-MRS we can analyse various metabolite ratios in the tissue, and obtain three different kinds of information. Energy metabolite ratios such as phosphocreatine (PCr) to adenosine-triphosphate (ATP)-PCr/ATP, PCr to inorganic phosphate (Pi)-PCr/Pi and Pi to ATP (Pi/ATP) are related to the energy pool. Second, phospholipide ratios such as phosphomonoesters (PME) to phosphodiesters (PDE)-PME/PDE provide the insight in cell membrane metabolism-its synthesis and degradation. Third, cerebral magnesium (Mg) and intracellular pH can be calculated [10].

In this prospective study, we explored differences in brain energy metabolism in primary GBMs with different MGMT and EGFR status, using 31-P-MRS.

## 2. Materials and Methods

### 2.1. Patients

Fifty-two patients who were admitted from 2016 to 2019 with high-grade glioma and received 31-P-MRS examinations were initially included. All patients underwent either the surgical resection of the tumor or biopsy.

### 2.2. MRI Acquisition

31-P-MR spectroscopy was performed on a 3T whole-body, multi-nuclear system (Skyra, Siemens Medical AG, Erlangen, Germany) with a double-tuned 1-H/31-P volume head coil (Rapid Biomedical, Würzburg, Germany). As described previously in [11], the axial layer arrangement was chosen to include as much brain volume as possible and as little as possible structures perturbing the 31-P spectra.

For each patient, a MRS 3D block of the whole brain, planned on a previously attained T2-SPACE sequence (sagittal-oriented, T2-weighted 3D sequence with isotropic resolution and a voxel size of 1.2 × 1.2 × 1.2 mm${}^{3}$ (TR = 3000 ms, TE = 412.0 ms, TA = 2:50)), was acquired via chemical shift imaging.

Similar to the process described by Hattingen et al. [12], the MRS acquisition was performed with WALTZ 4 proton decoupling, repetition time TR 2000 ms, echo time TE 2.3 ms, flip angle 60°, and 10 averaged acquisitions, in order to obtain a reasonable signal to noise ratio and also to cancel out possible movement artefacts during measurement, resulting in an acquisition time (TA) of 10:44 min.

### 2.3. Image Analysis

31P-MRS data analysis was processed offline with the jMRUI software package (version 5.0, http://www.mrui.uab.es, assesed on 1 March 2014.) using the non-linear least square fitting algorithm AMARES, which considers prior knowledge [13]. The fitting model was composed of 12 Lorentzian-shaped, exponentially decaying sinusoids as follows: phosphocholine, phosphoethanolamine (the sum of both referred to as phosphomonoesters or PME), inorganic phosphate (Pi), glycerophosphocholine, glycerophosphoethanolamine (the sum of both referred to as phosphodiesters or PDE), phosphocreatine (PCr), and adenosine triphosphate (ATP) consisting of two doublets (γ-ATP and α-ATP) and triplets (β-ATP), which were added together. The peak of β-ATP can be used as an internal quantification as it is considered uncontaminated by α-ADP, nicotinamide adenine dinucleotide NAD and NADH (oxidized) contributions to α-ATP [14]. In this study, we calculated the the means of α-ATP, β-ATP and γ-ATP as our defined reference value and designated it ATP. Additionally, intracellular pH was calculated using the chemical shift of Pi relative to PCr, using the formula of Petroff et al. [15].

The parameters of Mg and pH were calculated according to the formulas from Iotti et al. [16,17] as follows: the changes in Mg concentrations were estimated from the chemical shift difference between the ATPβ and the PCr signal (δβ).
(1)$$pMg=4.24-log_{10}\left[\frac{{\left(18.58+\Delta \beta \right)}^{0.42}}{{\left(-15.74-\Delta \beta \right)}^{0.84}}\right]$$

And the pH value was determined from the signal position of inorganic phosphate (δPi) regarding to PCr, set as the main reference. pH was than calculated according to the formula
(2)$$pH=6.706-0.0307\left[Mg\right]+log_{10}\left[\frac{\left(\Delta Pi-3.245\right)}{\left(5.778-\Delta Pi\right)}\right]$$

Absolute quantification of metabolite concentrations from 31P spectra was found to be inefficient within this study, due to possible limiting issues such as coil sensitivity, field inhomogeneity and scanning time for the patients. It has been shown that incorporating metabolite concentration ratios to evaluate metabolic changes is more stable [18].

The following two brain areas (contrast enhancing tumor voxels and contralateral area) were delineated by two experienced neuroradiologists and analysed with one or more 31-P-MRS voxels in each volunteer (see Figure 1). The amount of the included voxels per area depended on the size of the respective area and the quality of the spectra. CSF was excluded, only voxels which contained at least 2/3 of the investigated brain region were included. Please note that due to the voxel size gray and white matter could not be distinguished. Each single spectrum was assessed visually according to the criteria set forth by Kreis [19], and metabolite ratios were calculated afterwards. The ratios calculated were PCr/ATP, PCr/Pi, Pi/ATP and PME/PDE. Additionally, pH and Mg values were calculated.

### 2.4. Neuropathological Assessment

Neuropathological diagnosis including molecular markers of the tumor was routinely assessed using standard histopathology, immunohistochemistry, in situ hybridization, and sequencing technologies. For molecular classification, following markers, among others, were determined in routine analysis: IDH1 status, MGMT methylation status and EGFR amplification status. The MGMT methylation status was assessed by pyrosequencing using the Therascreen MGMT Pyro Kit (QIAGEN, Hilden, Germany). Cases with a mean methylation percentage of more than 8% were considered to be MGMT methylated [20]. Immunohistochemistry was performed to assess EGFR amplification status (clone 3C6; Ventana, Oro Valley, AZ, USA) and IDH1 mutation status (clone DIA-H09; Dianova, Hamburg, Germany) [21,22]. Presence of oligodendroglial component was excluded by 1p19q codeletion status.

Because of clinical and prognostic impact, in this study we focused only on MGMT and EGFR alterations in IDH1 wild type GBM.

### 2.5. Statistical Analysis

Statistical analysis was performed using R (R Core Team v. 3.6.1).

Data normality of metabolite ratios was assessed with the Shapiro–Wilk normality test and the one-sample Kolmogorov–Smirnov test at a 5% significance level. Quantile-comparison plots and histograms were presented. As the data are not normally distributed, a Mann–Whitney U-test was applied for the investigation of differences between groups, *p*-values < 0.05 were considered statistically significant. Boxplots were created for the visualisation of the results.

Survival analysis was done using Kaplan-Meier estimator.

As the best survival results were present when calculated for multiple molecuar alterations groups, we also undertook a between-group analysis with Kruskal-Wallis rank sum test. As post hoc test for multiple comparisons between groups we used pairwise comparisons with Wilcoxon rank sum test with Benjamini-Hochberg continuity correction.

### 2.6. Ethics Statement

This study was approved by the local Ethics Committee (AN 5100 325/4.19), and conducted in compliance with the Declaration of Helsinki.

## 3. Results

### 3.1. Patient Characteristics

Fifty-two patients with GBM were included in the study, 34 males (65.4%) and 18 females (34.6%). They had a median age of 67 years (range of 27 to 84). Patients with diagnoses other than primary GBM were excluded. In addition, patients with corrupted 31P-MRS spectrum were also excluded. To conclude, 43 patients with IDH1-wild type GBM were included in the study (Table 1).

### 3.2. Descriptive Statistics Regarding Molecular Markers

The largest group were patients with MGMT-unmethylated, EGFR amplified tumors and MGMT-methylated, EGFR amplified tumors. They were followed by patients with MGMT-unmethylated, EGFR not amplified tumors. The smallest sample of patients were those with MGMT-methylated, EGFR not amplified tumors (Table 2).

### 3.3. Survival Analysis

Based on survival analysis, our patients did not experience statistically significant differences in overall survival in regards to the MGMT methylation status (two-sided test *p* = 0.2, one-sided test *p* = 0.1). MGMT methylation status is usually not given only as methylated-unmethylated, but also with a degree of methylation. We used only methylated-unmethylated division, without exact degree of methylation, which is maybe why it resulted in no statistical significance in regard to the survival. Cox regression analysis showed that there is no statistically significant association of overall survival in MGMT with sex and age of the patients (*p* = 0.08). With a *p*-value of 0.9, no statistically significant difference in survival was observed between patients with amplified or not-amplified EGFR expression status.

Patients with MGMT-methylation and no EGFR-amplification showed slightly better survival than other clusters of patients, represented by the dotted purple curve in Figure 2. (*p* = 0.004).

### 3.4. 31-P-MRS Metabolites and Pathology Data Correlation

Whole sample analysis, as well as analysis by slices was performed (we used two slices for each patient).

EGFR amplified tumors had significantly lower PCr/ATP (*p* = 0.002) and PCr/Pi (*p* < 0.0001) ratios, and higher Pi/ATP (*p* = 0.006) and PME/PDE ratios (*p* < 0.0001). Patients with MGMT-methylated tumor had significantly higher Mg values (*p* = 0.01) and PME/PDE ratio (*p* < 0.0001), while their PCr/ATP (*p* = 0.03) and PCr/Pi (*p* = 0.04) ratios were lower (Figure 3).

### 3.5. Group Comparison

Next, the MRS metabolites were compared for four various combinations of molecular alterations in tumor voxels, and between tumor voxels and the contralateral (“healthy”) voxels of the same patients as control group.

#### 3.5.1. Intracellular pH

Intracellular pH was significantly higher in tumor voxels compared to the contralateral group in all molecular alteration groups as depicted in Figure 4. (*p* < 0.0001). There was no significant difference between various groups with respect to MGMT methylation and EGFR amplification status.

#### 3.5.2. Cerebral Magnesium Levels

Cerebral magnesium was significantly higher in patients with MGMT methylation and no EGFR amplification in tumor and contralateral side (*p* < 0.0001). Mg in tumor voxels of patients with MGMT methylation and EGFR amplification was significantly higher than in tumor voxels of patients with MGMT unmethylated tumors and no EGFR amplification with *p* = 0.02 (Figure 4).

#### 3.5.3. PCr/ATP Ratio

PCr/ATP ratio was significantly lower in tumor voxels than in contralateral control voxels in all groups except for MGMT methylated, EGFR not amplified group. Highest statistical significance was in MGMT methylated, EGFR amplified group (*p* < 0.0001).

Same group had significantly lower PCr/ATP ratio than all other groups, with highest significance in regards to the MGMT methylated, EGFR not amplified group (*p* = 0.00049).

#### 3.5.4. Pi/ATP Ratio

All groups had significantly higher Pi/ATP ratio in tumor voxels compared to contralateral side except for MGMT unmethylated, EGFR not amplified group. Highest statistical significance was observed in MGMT methylated, EGFR amplified group (*p* < 0.0001).

Pi/ATP ratio was significantly higher in patients with MGMT methylated, EGFR amplified tumors than in patients with MGMT unmethylated, EGFR not amplified tumors (*p* = 0.03).

#### 3.5.5. PCr/Pi Ratio

All groups except for MGMT unmethylated, EGFR not amplified group had significantly lower PCr/Pi ratio in tumor voxels compared to contralateral voxels (*p* < 0.0001).

Patients with MGMT methylated and EGFR amplified tumors had significantly lower PCr/Pi ratio when compared to MGMT methylated, EGFR not amplified group (*p* = 0.001), and to MGMT unmethylated, EGFR not amplified group (*p* = 0.0004).

Patients with MGMT unmethylated, EGFR amplified tumors had lower PCr/Pi ratio when compared to MGMT methylated, EGFR not amplified group (*p* = 0.02), and to MGMT unmethylated, EGFR not amplified group (*p* = 0.01).

#### 3.5.6. PME/PDE Ratio

All groups had significantly higher PME/PDE ratio in tumor voxels than in contralateral voxels (*p* < 0.0001) (Figure 5). MGMT methylated, EGFR amplified group had significantly higher PME/PDE ratio when compared to all other groups (*p* < 0.0001). Patients with MGMT unmethylated, EGFR amplified tumors had significantly higher PME/PDE ratio compared to MGMT methylated, EGFR not amplified group (*p* = 0.03).

## 4. Discussion

In this study, we aimed to identify differences in 31P-metabolites in genetically defined glioblastoma subtypes.

Various studies showed that imaging is capable in discovering differences in brain tumors regarding genetic status. Using proton magnetic resonance spectroscopy, Choi et al. [23] showed that 2-hydroxyglutarate (2HG) correlates with the presence of IDH1 and IDH2 mutations in gliomas. The authors speculated that this biomarker could also be used in follow-up diagnostics. Diehn et al. [24] demonstrated that the spatial distribution of gene expression within tumors could be depicted by integration of functional genomic datasets and medical imaging. The most important imaging marker was the ratio of contrast enhancing volume to the necrotic tumor volume, which correlated with EGFR expression. Another study by Young et al. showed correlation between restricted water diffusion and EGFR amplification [25]. Gupta et al. demonstrated higher relative cerebral blood volume in MR-perfusion in patients with higher degree of EGFR amplification [26].

Consequently, several MRI techniques were used in order to determine imaging differences regarding the genetic profile of the imaged tumors in vivo and in non-invasive way: proton spectroscopy, diffusion imaging, perfusion imaging. We chose 31-P-MRS as cell metabolism is severely altered in gliomas [27] and 31-P-MRS allows in vivo evaluation of energy metabolites [28]. New studies propose using inhibitors of oxidative phosphorylation such as gboxin [29] or metformin [30] in treatment of GBM. All these findings emphasize the importance of understanding energy metabolism of GBM. This could become especially important in future precision medicine approaches.

In our study cohort, 72.09% of the patients had EGFR-amplified GBMs. As we focused on IDH1-wild-type, or primary GBMs, this finding is in agreement with the literature which reports amplification of EGFR in around 60% of patients with primary GBMs [31]. Amplification of EGFR is commonly seen in brain tumors [32], making it a possible target for therapies. For this reason several EGFR inhibitors were developed, including targeted monoclonal antibodies, small molecules, as well as EGFR-kinase inhibitors. So far, all these attempts have failed [33]. In our study, no significant difference in overall survival was shown in patients with EGFR amplification and those without. Generally, EGFR amplification is considered to be a poor prognostic factor in GBMs, although studies on this topic are not uniform and with several significant limitations [34].

MGMT was methylated in 55.8% of our patients, a number comparable to the literature [35]. MGMT is a nuclear protein involved in a repair of alkylated DNA. Because of this characteristics, patients with MGMT-methylated GBMs respond better to the therapy with alkylated agents such as temozolomide. Our results are in line with the previous findings that patients with MGMT-methylated GBM live longer [7,8]; however, the significance level was not reached. When stratified with EGFR-amplification status, the results showed significantly higher survival in patients with MGMT-methylation and no EGFR-expression compared to other groups. The possible reason for this unexpected result regarding the MGMT-methylation and survival was explained earlier, and it is also given in the limitation of the study section. Using 31-P-MRS we analysed numerous cerebral metabolites in tumor voxels, and compared their occurrence between the “altered” and “non-altered” group. Patients with EGFR-amplified tumors had significantly lower PCr/ATP and PCr/Pi ratios, and higher Pi/ATP and PME/PDE ratios than patients without EGFR amplification. These results might indicate lower oxidative capacity of the EGFR amplified tumors.

The same could be said for the MGMT-methylated tumors. Patients with MGMT-methylated tumor had significantly higher Mg values and PME/PDE ratios, while their PCr/ATP and PCr/Pi ratios were lower than in patients without MGMT methylation.

Even though it was common belief that tumors have an acidic environment [36], MRS pH measurements revealed different results. For brain tumors it was shown that they are more alkaline than normal brain tissue [37,38]. Our results indicate that tumor tissue was more alkaline than the contralateral “healthy” brain tissue.

A study by Chandra et al. using secondary ion mass spectrometry showed increase in total Mg levels in infiltrating tumor cells compared to the normal brain tissue [39]). MR spectroscopy studies reported similar results [40], and also our study revealed increased cerebral Mg levels in MGMT-methylated tumors compared to those without the methylation.

The goal of a 31-P-MRS study is to detect and accurately measure phosphorous metabolites in vivo. However, a spectrum measured by 31-P-MRS is considered to depict relative concentration of metabolites in a voxel [41]. Consequently, the ratios instead of the absolute values of the metabolites are frequently reported.

Phosphocreatine (PCr) is the energy buffer that prevents rapid fluctuations in ATP, the energy currency of the cell [42]. Their ratio acts as an important marker of high-energy phosphorous metabolism in the brain [43,44]. Studies showed that the PCr/ATP ratios are lower in high grade gliomas than in low grade ones [45,46]. Amplification of EGFR, which is proven in numerous tumors leads to the activation of proliferation and cell survival signals [47]. It is well documented that the cells with higher proliferation rate have to rewire their metabolic profile in order to acquire enough energy during this process. For example, cells stimulated with epidermal growth factor often show increased ability to take up nutrients [48]. PCr/ATP levels were significantly lower in MGMT methylated compared to those with unmethylated MGMT, as well as in EGFR amplified tumors compared to those without EGFR amplification. PCr/ATP ratio was lower in MGMT methylated, EGFR amplified tumors when compared to all other mutation combinations. Increased cytosolic ATP is observed in apoptotic cells [49]. Our results could indicate higher apoptotic rate in EGFR amplified GBMs than in those without this molecular trait. This could also indicate a higher apoptotic activity in tumor cells with EGFR-amplification regardless of the MGMT-methylation status. It is also possible that the rewiring and adapting of the energy metabolism could be behind this change in PCr/ATP ratio. Of course, as prevously explained we could not measure the individual metabolites but only their ratio, so it is impossible to say which metabolite is “to blame” for the change of the ratio. Furthermore, it is impossible to say if these traits are intrinsic to the tumor or maybe the consequence of the therapy.

Inorganic phosphate (Pi) is a very important metabolite in the ATP formation, kinase/phosphase signalling, and synthesis of other biochemical products [50]. In muscle, PCr/Pi ratio is a marker of mitochondrial function [51].

Mitochondrial dysfunction is a well-known feature of multiple tumors, including gliomas. It includes preference of glycolysis over oxidative phosphorylation, mitochondria mediated apoptosis, and enhanced reactive oxygen species generation [52]. In ATP catalysis, the ADP, energy, and Pi is created, hence the Pi/ATP ratio could be perceived as a marker for the amount of ATP turnover.

The PCr/Pi ratio is deduced to correlate positively with the phosphorylation potential [53] or the tissue oxygenation [54] and can be interpreted as a marker of metabolic oxidative capacity [55,56]. Our results suggest the decreased PCr/Pi ratio in patients with EGFR amplified or MGMT methylated tumor. In group comparison analysis, this ratio was lower in patients with MGMT methylation and EGFR amplification when compared to patients with MGMT methylated, EGFR not amplified tumors, and MGMT unmethylated, EGFR not amplified tumors. Decreased PCr/Pi ratio in MGMT-methylated and EGFR-amplified tumors could indicate reduced energetic state or tissue oxygenation due to the increased necrosis, similar as that stipulated in [11].

Phosphomonoesters (PME) include phosphocholine and phosphoethanolamine. Phosphocholine is intermediate in phosphatidylcholine synthesis [57] and phosphoetanolamine together with phosphatidylcholine constitutes the most abundant phospholipid in all mammalian cell membranes [58]. Therefore, PME are of utmost importance to the membrane structure. Phosphodiesters (PDE) which include glycerophosphocholine, glycerophosphoethanolamine and mobile phospholipids, are cell membrane breakdown products [37]. PME/PDE ratio can therefore be considered to be a cell membrane metabolism marker [59]. Cell membrane metabolism changes are well known in gliomas. Since the 1950s it was known that brains with glioma have elevated lipid levels [60,61]. Elevated PDE concentration is associated with decreased membrane turnover [62]. Several studies showed altered PME/PDE ratio indicates change in various pathological states. For example, decreased brain PME/PDE ratio in bipolar disorder is linked with differences in membrane turnover [29], increased PME/PDE ratio was found in astrocytomas, lymphomas and metastases compared to normal brain [37]. Our results show increased PME/PDE ratio in tumor voxels when compared to contralateral side. Increased PME/PDE ratio was found in MGMT methylated, EGFR amplified group in comparison to all other groups of molecular alterations. Altered membrane metabolism is well known feature of GBM [11]. Here we show the possible differences in membrane metabolism between tumors with different MGMT and EGFR status. PME/PDE ratio together with PCr/ATP and PCr/Pi ratios give us an insight in cell reproduction rates and tumor growth [37]. These results could reflect faster cell reproduction and tumor growth in MGMT-methylated and EGFR-amplified tumors.

## 5. Limitations of the Study

MGMT methylation status is usually not given only as methylated-unmethylated, but also with a degree of methylation (unmethylated, under cutoff, methylated 10–29%, methylated 30–100%) [63]. We divided the patients only in methylated and unmethylated groups, which is maybe the reason it resulted in no significant difference in overall survival between the two groups.

All patients with suspicion on GBM received full pallet of molecular markers. In this paper, we limited ourselves on the MGMT and EGFR analyses, in particular because of their clinical importance. For other markers we did not have sufficient diversity of data (for example only three patients had negative p53 mutation, and only two of them had ATRX loss).

## 6. Conclusions

This study used 31-P-MRS metabolites measurement and correlated those with molecular features of GBM. Brain energy metabolites were different based on MGMT and EGFR status of the tumors, which could indicate variances in biochemistry in these subtypes of GBM. We are still far away from completely understanding tumor biology of GBM. This study could help us better stratify different subclasses of GBM further pointing out differences between tumors with distinct molecular footprint.

## Figures and Tables

**Figure 1 cancers-13-03569-f001:**
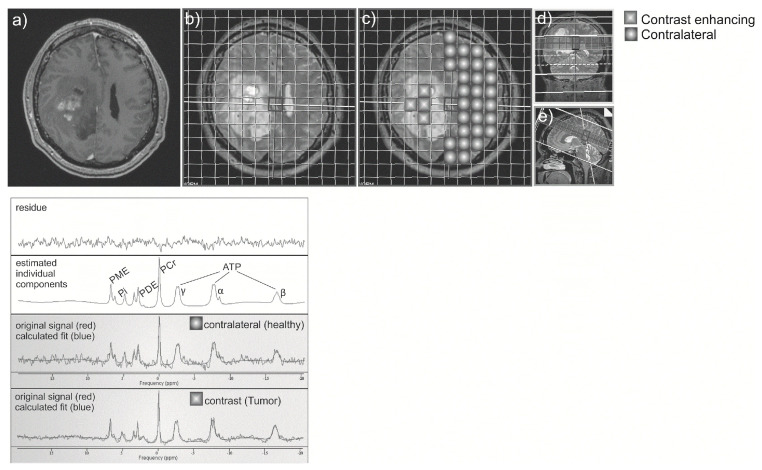
Contrast-enhanced T1 axial image (**a**) was registered to the spectroscopy-grid (**b**), and two regions: contrast-enhancing or tumor region, and contralateral “healthy” region were chosen (**c**). Correct position of voxels was checked on coronal (**d**) and sagittal (**e**) T2 images. Spectroscopy curve was depicted for each voxel (**below**).

**Figure 2 cancers-13-03569-f002:**
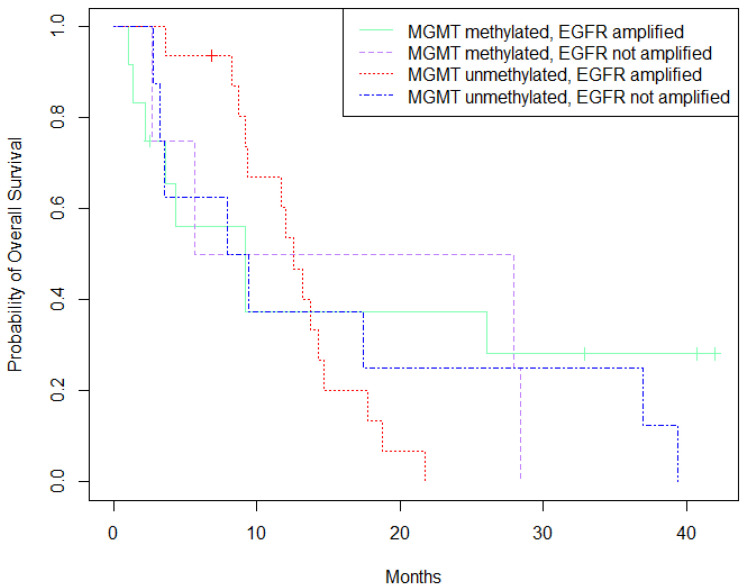
Survival analysis (Kaplan-Meier) based on MGMT-methylation and EGFR-expression status.

**Figure 3 cancers-13-03569-f003:**
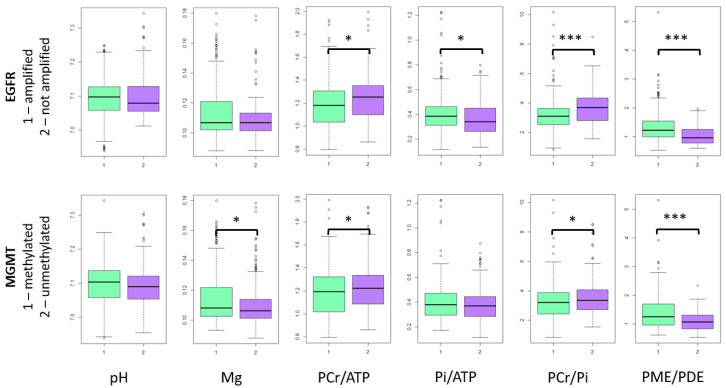
31-P-MRS metabolites in tumor voxels. In the figure only whole sample results are presented, because slice results are mainly the same. *—*p* < 0.05, ***—*p* < 0.0001.

**Figure 4 cancers-13-03569-f004:**
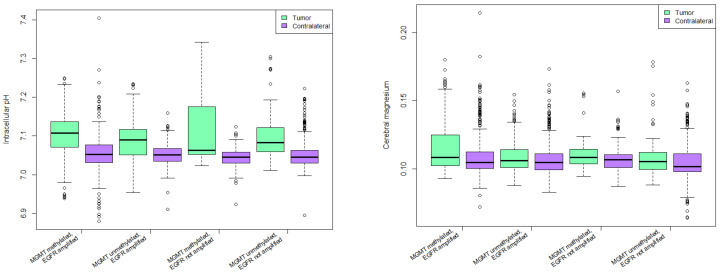
Intracellular pH (**left**) and cerebral magnesium (**right**) in various combinations of molecular alterations.

**Figure 5 cancers-13-03569-f005:**
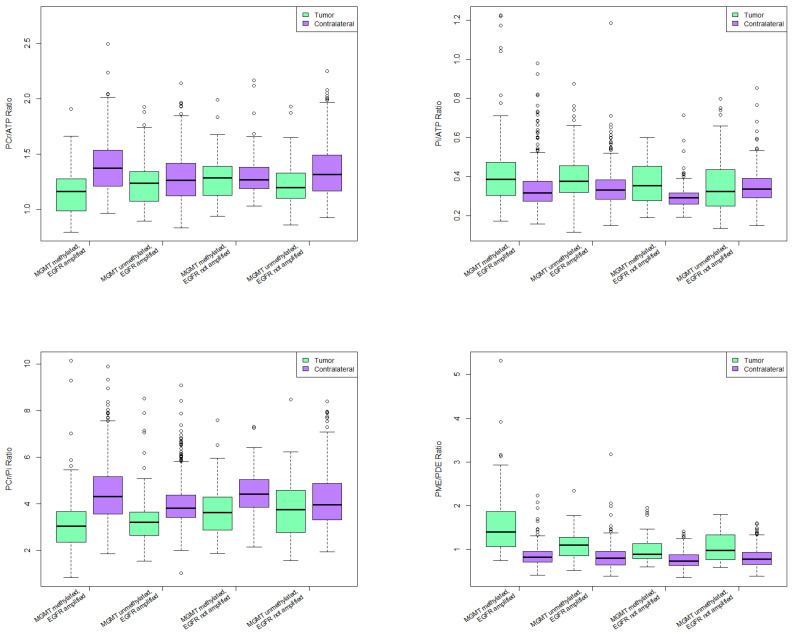
31-P-MRS ratios in various combinations of molecular alterations.

**Table 1 cancers-13-03569-t001:** Patient characteristics.

**Total patients with GBM**	**52**
Males	34 (65.4%)
Females	18 (34.6%)
**Age (in years) at the time of diagnosis**	
Median age	67
Range	27–84
**Final study numbers**	**n**
GBM excluded due to the IDH1 positivity	4
GBM excluded due to the bad quality of 31-P-MRS spectrum	5
**Total IDH1-wildtype GBM patients included in the study**	**43**

**Table 2 cancers-13-03569-t002:** Number of patients in each category.

	**Amplified**	**Not amplified**	**Not available**
**EGFR**	31 (72.09%)	12 (27.9%)	0 (0%)
	**Unmethylated**	**Methylated**	**Not available**
**MGMT**	24 (55.8%)	18 (41.8%)	1 (2.3%)
MGMT unmethylated, EGFR amplified			17 (40.4%)
MGMT methylated, EGFR amplified			13 (30.9%)
MGMT unmethylated, EGFR not amplified			7 (16.6%)
MGMT methylated, EGFR not amplified			5 (11.9%)

## Data Availability

Data will be made available upon request to the corresponding author.

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
