# Peer review of "Phosphorous Magnetic Resonance Spectroscopy and Molecular Markers in IDH1 Wild Type Glioblastoma"

_cancers, 2021, doi:10.3390/cancers13143569_

Round 1

Reviewer 1 Report

The article entitled: Spectroscopy Imaging and Molecular Markers in IDH-wild-type Glioblastoma”, is based on the use of 31-P-MRS to explore differences in brain energy metabolism in 50 GBMs with different MGMT and EGFR status, focusing on parameters such a levels of various phosphorous metabolites in vivo.

Overall the topic addressed by the authors is extremely important. Imaging parameters have potentially crucial significance if analyzed with clinical status or as in this study with molecular subtype. In the GB pathology where we are still so far from a real understanding of tumor biology, from the development of effective therapies, and where there is still difficulties in stratifying subclasses, it is crucial to explore anything that could give clues for better stratification.  Therefore I believe that this study will enrich the glioblastoma scientific community. The paper is acceptable but I absolutely recommend a series of minor revisions.

Comments:

1)            The word O6-Methylguanin-DNS-Methyltransferase has to be corrected: DNS is DNA

2)            The introduction should explain better the importance and the use of the various parameters that will be measured. The ratios calculated must be explained for significance

3)            “Tumor voxels” should be explained as a term. Moreover is not always used in the paper, please specify the meaning.

4)            How was it measured the EGFR amplification and MGMT methylation status?

5)            Table 2 should be relisted, it would be clearer to list the different populations in the order of most frequent.

6)            Figure 2: legends should be more explicative, instead of 1 or 2 for an easier and faster understanding should be extended to what they refer to in an extended way.

7)            Figures are never mentioned in the text.

8)            Cox regression is it somewhere reported?

9)            In the paragraph on page 5 line 150 and 151, to which curve are the authors referring?

10)         Line 148, it is not correct the term EGFR positive or negative status, it should always be amplified and not amplified.

11)         Line 162, genetic mutations? Not the correct word. Better molecular alterations. Same thing in line 258.

12)         The whole discussion needs to be revisited thoroughly. All results are reported but not discussed, it would be interesting if authors give interpretations of what is going on, and why some metabolites or parameters are changing between normal and tumor tissues and between tumors with a different genetic profile

Reviewer 2 Report

Summary:

This paper aims to find differences in subtypes of Glioblastomas (GBM) by 31P MRS for potential therapeutic strategies.  

Comments:

Abstract: Not all abbreviations are defined.

Introduction:

The aim of this article is to look at 31P biomarker of different GBM subtypes more discussion of the actual relevance of 31P MRS in GBM is warranted. Currently there is only 3 lines talking about 31P MRS in the introduction. What information will the different metabolites give you and how is this significant for the tumor as biomarkers and potential treatments? What and why are different ratios used.

Line 26- 27: Abbreviations are not defined when they are first mentioned or not defined at all

Line 29-31. Explanation is confusing

Materials and Methods:

Line 59: It seems like Chemical Shift Imaging sequence is used? If so it should be described as such. Why use this approach and not a single voxel technique such as ISIS (I realize that this is not standard on Siemens scanners but there are WIPs and ways to acquire SVS 31P). With SVS techniques, it should be much easier and potentially faster to acquire a select voxel in tumor and corresponding contralateral side without motion or phase issues with CSI techniques.

Line 67: How long was the scan? How was motion mitigated if the scan is long?

Line 69: What if any pre-processing was done on each individual voxel. Are they phase corrected individually or automatically with the jMRUI software. If not did this affect the quantification of the metabolites.

Line 84-86: If this is so, what are you reporting in Results 4.5.3 (line 175)

Line 92: What is the target ROI?

Figure 1: The current figure is informative but representative spectra from healthy, and different GBM tissue should be included to and complement this figure. This figure looks Very similar to a figure in a publication by Lisa Maria Walchhofer et al 1 which begs the question if the data presented in the manuscript has been published before?

Line 125: Two “Results” headlines

Line 129: What determined if a spectrum was corrupted and on what criteria was it excluded?

Line 169: How are Magnesium level quantified and how is this significant in your study? There is no description in methods section of Mg quantification

Figure 4-5. Are there no significant difference between the groups in these figures? Figures are also not referenced to in the text which makes it har to relate especially with the limited explanation in the some of the figure legends.

Line 175: How is total brain ATP determined? If there re partial volumes effects, e.g. there are healthy tissue or as you mentioned in the methods that “at least 2/3 of target ROI is included” meaning are there partial volume effects affecting your ATP quantification?

Line 258-260: What does this mean biologically

General: Not all abbreviations are defined as they are first mentioned
